# A High-Temperature-Resistant Stealth Bandpass/Bandstop-Switchable Frequency Selective Metasurface

**DOI:** 10.3390/mi15080948

**Published:** 2024-07-24

**Authors:** Gengyuan Bao, Peng Li, Jing Sun, Erzhan Chen, Shaojie Li

**Affiliations:** State Key Laboratory of Electromechanical Integrated Manufacturing of High-Performance Electronic Equipments, Xidian University, Xi’an 710071, China; m15203802038@163.com (G.B.); 23041212814@stu.xidian.edu.cn (J.S.); m18917039739@163.com (E.C.); 23041212827@stu.xidian.edu.cn (S.L.)

**Keywords:** liquid metal, frequency-selective surface, electrically reconfigurable, thermal, low radar cross-section

## Abstract

We propose a bandpass/bandstop-switchable frequency-selective metasurface (FSM) designed for high-speed vehicles that generate high temperatures during flight, based on a high-temperature-resistant dielectric substrate and liquid metal (LM). We fabricated a cavity structure by utilizing a high-temperature-resistant dielectric substrate to form a metal FSM element by introducing LM, enabling specific electromagnetic functions. The flow state of the LM can be controlled to achieve the switching of the FSM’s bandpass/bandstop performance. The bandstop characteristic has a resonance frequency of 6.1 GHz and the bandpass interval is 5.53–6.51 GHz. The bandpass characteristic has a resonance frequency of 5.41 GHz and the bandstop interval is 5.30–5.76 GHz, achieving a bandpass/bandstop switching range of 5.53–5.76 GHz. LM fluidity can aid in high-temperature heat dissipation. When the LM reaches a certain flow rate, the FSM structure’s average temperature can be reduced by an order of magnitude from a thousand to less than a hundred degrees. The FSM exhibits low RCS, with 22.35 dB and 36.79 dB reductions in bandstop and bandpass properties, respectively, compared with that of sheet metal. A prototype was developed and tested, validating the design of the FSM structure with high-temperature resistance, bandpass/bandstop switchability, and low RCS characteristics, and is expected to be applied in high-speed aircraft.

## 1. Introduction

A frequency-selective metasurface (FSM) is a spatial filter structure comprising a two-dimensional periodic arrangement of metallic resonant elements that selectively transmit or reflect electromagnetic waves irradiated onto them. These structures are commonly used as radomes with special functional requirements [1].

With technological advances, the operational environments and functional demands of FSMs are becoming increasingly complex. Satisfying the actual demands of a single-function frequency-selective surface (FSS) is becoming increasingly challenging. Therefore, numerous studies on electrically reconfigurable FSMs are underway [2,3,4,5,6]. Currently, reconfigurable technologies are categorized into mechanical and electrical methods [7].

Mechanical methods involve altering the dimensional structure of the FSM periodic element through external mechanical forces to modify the electrical properties. Cui et al. utilized 3D printing to design an origami-type FSM with a photopolymer rubber-like elastomer as the base material. This design changes the interaction between the incident field and conductive units through stretching and deformation, resulting in a shift in resonance frequency [8]. Nauroze et al. developed a reconfigurable FSM with thermotropic properties using a polyester film with thermal sensitivity to the folding state by controlling the change in the external temperature [9]. Additionally, Chen et al. designed a 3D rotationally reconfigurable FSM that controls the angle of the ring fixed above the linkage, to alter the incidence angle of electromagnetic waves. This was achieved by rotating the driving rod to achieve a small dynamic tenability [10]. The mechanical reconfiguration of electrical properties was slow, the accuracy was susceptible to errors, and the control was complex.

The utilization of electronic control methods is essential for achieving the reconstruction of electrical properties through electronic control devices, such as PIN diodes, variable capacitance diodes, and microelectromechanical system (MEMS) switches [11,12,13]. Che et al. proposed an FSM that loads switching diodes between the metal slots of the transmission layer, and the transmission near 3.7 GHz can be switched by controlling switching the on/off state of the diodes [14]. Bouslama proposed an FSM featuring a tapered PIN diode intersected by a rectangular structure, where the switching state of the PIN diode was adjusted, allowing for frequency switching between 1.45 GHz and 2.45 GHz [15]. Mojtaba et al. proposed a tunable FSM based on a slotted grounded structure, achieving frequency shifts of over 1.7 GHz in the X-band by tuning a metallic MEMS bridge loaded on the slot [16]. However, these applications often necessitate additional feeder networks or a large number of MEMS switches, resulting in increased costs.

Among the various properties of FSM radomes, stealth characteristics are particularly intriguing [17]. The target radar cross-section (RCS) is influenced by numerous factors. Loading the FSM structure on the inside of the radome is a widely used method. Liao utilized an FSM of crossed elements on the radome dielectric substrate, strategically setting the substrate thickness to an odd multiple of a quarter wavelength to achieve phase cancellation in the two-layer structure and reduce return loss, resulting in a 25% RCS reduction in bandwidth [18]. Ji designed an absorptive transmissive FSM to achieve an insertion loss of less than 1 dB in the 8.5–10 GHz range and reduce RCS by over 15 dB in both the out-of-band 4.8–8.5 GHz and in-band 8.5–10.5 GHz bands [19].

Finally, the surface of high-speed vehicles generates excessive heat owing to friction with air during high-speed flight, which necessitates improved heat resistance for the FSM structure loaded onto it. Currently, ceramic-based wave-transparent materials are the preferred choice for developing high-temperature-resistant radomes [20]. While the utilization of liquid metals (LMs) with excellent thermal conductivity and fluidity to achieve the high-temperature characteristics of FSMs is still in its infancy, FSMs based on ceramic substrates and LMs show significant potential for heat transfer in high-speed aircraft and electromagnetic devices [21].

Therefore, this study proposes a novel FSM-performance reconfigurable approach by utilizing flowable LMs in place of a portion of the FSM element. By changing the morphology of the FSM element through the introduction and removal of LM, enhanced performance could be achieved.

## 2. Design and Analysis

### 2.1. FSM Element Design

First, the bandpass/bandstop 2D metal patterns were introduced as a novel approach to achieve the desired functionality. In this study, the circular slit shown in Figure 1a was proposed as a bandstop 2D pattern. The combination of the metal circular patch and ring formed a concentric circular slit structure. Additionally, another metal patch was incorporated into the periphery of this bandstop 2D pattern to form a second circular slit. Together, these elements formed a bandpass double circular slit structure, as shown in Figure 1b.

The aforementioned two-dimensional pattern was stretched longitudinally to form a three-dimensional FSM basic element. The peripheral metal portion from Figure 1b can be considered a unified entity when the FSM basic elements are arranged periodically. This portion could replace the metal component owing to the fluidity of the LM. The structure of the proposed bandpass/bandstop-switchable 3D FSM element is shown in Figure 2. To maintain concentricity between the metal ring and circular patch during processing, the circular patch and metal ring were processed on a Rogers RT/Duroid 5880 dielectric material (with a permittivity of 2.2 and loss tangent of 0.0009), known for its stability. The metal rings and circular patches were loaded onto the discs using printed circuit boards (PCBs) to ensure the properties and feasibility of the FSM process, as shown in Figure 2a. The aforementioned structure was encapsulated in the center of a polymethylmethacrylate pillar (PMMA, with a permittivity of 2.7 and loss tangent of 0.0078) to form the structure shown in Figure 2b. A substrate was added underneath and on top of the PMMA pillar to form the element structure. The parameters related to the materials used in this structure are shown in Table 1.

The LM discussed in this study was EGaIn, comprising gallium (68.5%), indium (21.5%), and tin (10%). EGaIn exhibited excellent electrical and thermal conductivity at room temperature, with PMMA utilized as the dielectric substrate material. When multiple elements were periodically arranged, multiple pillars as well as upper and lower substrates formed a cavity structure that served as the LM flow channel. When no liquid metal was introduced into the cavity structure, a fixed circular gap was formed between the metal ring encapsulated in the circular pillar and circular patch, achieving a bandstop characteristic (hereinafter referred to as Case I), as shown in Figure 2c. When the LM was introduced into the cavity, a second uniform circular gap was formed between the metal ring encapsulated in the pillar and the LM owing to the isolation effect of the circular pillar, resulting in a bandpass characteristic (hereinafter referred to as Case II), as shown in Figure 2d.

The parameters of the bandpass/bandstop-switchable FSM element are shown in Figure 3, and the specific structural parameters are listed in Table 2. These parameters included the element period P = 15 mm, circular patch radius R1 = 4.56 mm, first circular gap width W1 = 0.21 mm, metal circular width W2 = 0.8 mm, second circular gap width W3 = 0.8 mm, cylindrical pillar radius R2 = 6.44 mm, LM flow channel height H = 2 mm (cylindrical pillar height), upper/lower dielectric substrate thickness HB = 1.5 mm, processed disc radius R3 = 6.1 mm, disc height H1 = 2.5 mm, and disc thickness H2 = 0.2 mm.

The electrical properties of the FSM structure were simulated using HFSS, defining the master–slave boundary conditions and Floquet ports employed to simulate infinite periodic planar arrays. When the electromagnetic wave’s incident was vertical and the structure was in Case I, the S-parameter curve of the bandpass/bandstop-switchable FSM structure was as shown in Figure 4a. The working condition of the bandstop resonance frequency was 6.32 GHz, with an insertion loss (S21) of −22.40 dB and return loss (S11) of −0.61 dB, indicating good transmission performance. When the FSM structure was in Case II, the S-parameter curve was as shown in Figure 4b, with a bandpass resonance frequency located at 5.57 GHz, an insertion loss (S21) of −0.60 dB, and a return loss (S11) of −23.37 dB, also demonstrating excellent transmission performance.

By analyzing the transmission coefficients of the FSM structure in both cases, the transmittance of each case could be calculated. This information allowed for a better understanding of the bandpass/bandstop switching frequency range and bandwidth. The transmittance curves are shown in Figure 5. In Case I, the bandstop was identified as the interval in which the transmittance was less than 15%, that is, 5.53–6.51 GHz (the orange interval in the figure). In Case II, the bandpass interval was defined as the range in which the transmittance was greater than 85%, that is, 5.30–5.76 GHz (the green interval in the figure). The overlapping section of these intervals represents the range in which the FSM structure achieved the bandpass/bandstop-switchable function. The bandpass/bandstop-switchable FSM successfully operated within the 5.53–5.76 GHz region, showcasing excellent transmission performance for both functions.

### 2.2. FSM Stealth Property Analysis

The stealth performance of the proposed bandpass/bandstop-switchable FSM structure was simulated and analyzed for two cases. The RCS was numerically represented as the ratio of the power reflected from a target per unit cubic angle in the direction of the radar receiving antenna to the power density incident at the target. First, the RCS performance of the FSM structure was simulated using CST Studio Suite 2021 simulation software for both cases. Finally, the RCS curtailment rate was calculated by comparing the RCS of the corresponding frequency of an equal-sized metal plate as the reference value. The average value of the RCS within the range of ±0.5 GHz near the minimum RCS frequency of the sample was considered as the minimum value of the RCS. The results were classified into three types: (1) the RCS variation with frequency; (2) the RCS variation with the angle when the electromagnetic wave was TE-polarized at a frequency corresponding to the lowest RCS point; and (3) the RCS variation with the angle when the electromagnetic wave was TM-polarized. The RCS simulation results for the bandpass/bandstop-switchable FSM structure are shown in Figure 6, with specific data listed in Table 3.

For Case 1, the RCS minimum for the bandpass/bandstop-switchable FSM structure was at 5.8 GHz, with a minimum RCS value of −2.571 dBsm. At this frequency, the metal plate had an RCS value of 19.774 dBsm, resulting in a reduction of 22.345 dBsm compared with that of the equivalent-sized metal plate RCS. In Case II, the minimum RCS value was at 5.67 GHz, with a value of −17.25 dBsm. The RCS of the equal-sized metal plate at this frequency was 19.544 dBsm, representing a reduction of 36.794 dBsm compared with the RCS of the equal-sized metal plate.

When transverse electric (TE) polarization was observed, as shown in Figure 6b, both Cases I and II achieved full-angle RCS reduction. Case II primarily operated within the range of ±90°, with its lateral RCS reduction slightly lower than that of Case I, which was closer to that of the metal plate. However, its frontal RCS reduction surpassed that of Case I, with a reduction of 36.794 dBsm. When transverse magnetic (TM) polarization was observed, as shown in Figure 6c, Case I still achieved a full-angle reduction in RCS. However, Case II exhibited a cyclic increase and decrease in RCS, with a higher frontal RCS reduction compared with that of the metal plate, showing an increase of 36.794 dB with an offset angle. The frontal RCS reduction of Case II surpassed that of the metal plate, with the RCS values gradually converging to −15 dBsm as the offset angle increased. Notably, a significant RCS reduction was only realized within the range of ±30°. Transitioning from Case I to Case II, the RCS reduction rate in Case II increased at the front and decreased at the side.

Therefore, the FSM structure enhanced forward stealth performance by allowing the passage of LM, whereas withdrawing the LM enhanced side stealth performance. The stealth performances in the two cases differed, with Case 1 exhibiting wider RCS frequency and angle reduction intervals compared with those of Case II. However, the RCS reduction rate of Case II was higher than that of Case I. In terms of angle reduction, Case I aligned more closely with front-, side-, and full-angle RCS reduction, whereas Case II primarily focused on single-angle RCS reduction.

In summary, the stealth performance of the bandpass/bandstop-switchable FSM structure was characterized by high stealth performance at multiple angles within a narrow frequency band. This structure could be seamlessly switched between the two operating conditions to achieve specific stealth function requirements.

### 2.3. Thermal Analysis

The thermal simulation software COMSOL Multiphysics 6.1 was used to simulate the two cases of the structure. The 7 × 7 FSM elements were arranged in a two-dimensional periodic structure with equal spacing. The side boundaries around the structure were closed to form a cavity structure with a constant spatial extent. The two ports on opposite axes were set as the entrance and exit points for LM and air.

A 1000 W planar heat source was placed on the bottom surface of the FSM structure. Air and LM were used as cavity fluids to cool the device, with their relevant thermal properties listed in Table 4. The air inlet flow rate was set to 0.1 m/s, whereas the LM flow rate was set to 0.1 m/s, 0.2 m/s, and 0.3 m/s for comparison. The average and maximum temperatures on the top surface of the FSM structure were employed to evaluate the thermal performance of the FSM structure.

Based on the data in Table 5, the cooling and heat dissipation achieved using LM surpassed that of the air heat dissipation. The average temperature of the upper surface of the FSM structure with air heat dissipation could reach 1570.8 °C, whereas the utilization of LM decreased this value to 91.961 °C, showcasing the superior heat dissipation capabilities of LM. Furthermore, increasing the LM flow rate resulted in an improved heat dissipation performance. At an LM flow rate of 0.2 m/s, the average temperature dropped to 55.625 °C, and at 0.3 m/s, it further decreased to 43.423 °C. The utilization of LM allowed the average temperature of the upper surface of the FSM structure to reach 1570.8 °C.

The results are shown in Figure 7, illustrating that the temperature frequency was generally lower on the left side, whereas on the right side, the temperature was lower only at the exit. This suggests that multiple exit channels can be established on the right side to enhance the overall heat dissipation performance of the FSM structure.

In summary, the LM-based switchable FSM structure with a bandpass/bandstop demonstrated excellent heat dissipation capabilities, satisfying the high-temperature resistance requirements of the FSM structure in the radomes of high-speed aircraft.

## 3. Experimental Measurements and Discussion

### 3.1. Bandpass/Bandstop Measurement

The proposed bandpass/bandstop-switchable FSM structure was machined into a 375 mm × 375 mm × 5 mm test sample (25 × 25 elements) with a thickness of 2 mm around the outer wall. The metal rings and circular patches were composed of copper as the PCB metal material, machined on circular discs composed of Rogers RT/duroid 5880. The upper and lower substrates, along with the pillars, were composed of PMMA with optical transmittance. The prototypes of the LM-filled and LM-withdrawn FSM structures are shown in Figure 8.

Utilizing a vector network analyzer, the insertion loss S21 of the FSM prototype was measured and recorded under two scenarios. The transmission coefficient of the double-ridged horn antenna without the FSM prototype served as the error benchmark. By comparing the measured S21 with the benchmark, the S21 of the FSM prototype was determined and compared with that obtained from the simulation results. Experimental measured S21 as shown in Figure 9.

In Case I, the lowest resonance frequencies of the simulated and measured S21 curves were 6.32 GHz and 6.45 GHz, respectively. The maximum frequency shift was 2.05%, the offset was 0.13 GHz, and the minimum insertion loss values were −22.48 dB and −23.72 dB. In Case II, the lowest resonance frequencies of the simulated and measured S21 curves were 6.12 GHz and 6.25 GHz, respectively. The maximum frequency shift was 2.12%, the offset was 0.12 GHz, and the minimum insertion loss values were −32.05 dB and 34.85 dB. The shift in the experimental and simulation frequency points was attributed to three primary reasons as follows:During the transition of the FSM structure, the LM was not completely withdrawn, and residual LM droplets adhered to structures, such as pillars, causing a change in the metal pattern of the FSM structure and resulting in a rightward shift of the frequency point.In the LM filling process, obstructions such as pillars can lead to the formation of a portion of the bubble cavity, as shown in Figure 10. This resulted in the second circular gap of the working condition, and two double circular gaps were not fully formed, resulting in the shift of the frequency point to the right.Errors in the processing of FSM prototypes, such as the machining tolerances of structures and the imperfect concentricity of circular patches and metal rings, can also contribute to frequency point shifts.

### 3.2. RCS Measurement

The proposed FSM structure was machined into a 375 mm × 375 mm × 5 mm test specimen (25 × 25 elements) and sealed around the perimeter with a 2 mm thick dielectric substrate material.

To ensure consistency in positioning during multiple experiments, the prototype and fixture were placed in the center of the rotary table. The RCS variation of the bandpass/bandstop-switchable FSM structure was tested in a microwave anechoic chamber within the frequency range of 5–7 GHz. Because the bandpass/bandstop-switchable FSM structures exhibited symmetry in both forward and reverse directions, only the RCS variation of the FSM in the ±90°angle range at 6 GHz was measured. The electromagnetic waves used in the experiment were TE-polarized, and the curve trend of the simulation and measurement aligned closely, as shown in Figure 11.

For the bandpass/bandstop-switchable FSM structure, the RCS minimum point corresponded to a frequency of 5.85 GHz, as shown in Figure 11a, with a measured and simulated frequency offset rate of 0.86%. The RCS minimum value was 1.28 dB, an improvement from the simulation result of −2.57 dB. In the ±90° angle range, the measured peak in front of the prototype exhibited an angular offset phenomenon, the offset angle was −1.2°, and the lateral angular offset was 2.4°. The peak angle offset in the forward direction can be attributed to two main reasons:The FSM structure had an LM residue inside the sample during case changeover, and the RCS performance was significantly influenced by the metal material, resulting in an angular shift.The rotation of the test table caused the sample to tilt, leading to deviations in RCS measurements.

In conclusion, the trend observed in both the measured and simulated results of the prototype was consistent. Although offset phenomena were observed at special frequencies and points, the offset rate was controlled to be less than 1%. Additionally, the phase offset of the front and side during the angle test remained within 2°, and the RCS value remained low even in the event of an increase in RCS in the side direction. The main discrepancy between the experimental and simulation results can be attributed to the experimental and processing conditions. The experimental results unequivocally demonstrated the exceptional stealth performance of the two structures.

## 4. Conclusions

This study designed a stealthy bandpass/bandstop-switchable FSM based on LM. A comparison of the bandpass/bandstop-switchable FSM proposed in this study with other electrically reconfigurable FSMs from previous studies is shown in Table 6. In [22], a novel multipolarizable reconfigurable patch antenna was proposed to achieve multiple polarizations of the antenna by controlling the motion of LM in a 3D-printed microfluidic channel. A dual-layer dielectric cavity FSM that achieved frequency reconfigurability by switching the LM injection layer was proposed [21]. In [23], the spaced filling of mercury and mineral oil into a polytetrafluoroethylene tube was performed, and the pillar of mercury was controlled using a pressure-regulating device length to achieve dynamic tuning of the resonant frequency. In [24], the reconfigurable properties of absorbed/reflected electromagnetic waves were obtained by controlling the filling of water on the FSM of the bandpass type. In [25], dielectric elastomers were utilized as a substrate material, altering the size of the conductive cells through the application of high-voltage direct current to achieve electrical reconfiguration. In [26], a cross-tapered dipole was inserted between cross-tapered dipoles and the transmission line via a switch to achieve ultra-wideband bandpass/bandstop reconfigurable properties. In [27], LM was injected into FSM elastomers and switched from all-pass to bandpass/bandstop characteristics by controlling the flow of LM in microfluidic channels.

None of the aforementioned studies on electrical reconfigurability considered RCS reduction and heat dissipation performance. Therefore, the bandpass/bandstop FSM proposed in this study showed significant promise in addressing challenges related to stealth and high-speed flights in the future. Its advantages and some outlooks are summarized as follows:It easily switched between bandpass and bandstop in the C-band.It combined good frequency selection and reconfiguration performance, good heat dissipation performance, and RCS reduction performance. Therefore, it has great potential to cope with the future stealth and ultra-high-speed vehicle fields.It realized the function switching in the 200 MHz interval, but for some wideband antennas, this FSM cannot meet the working requirements, and the interlayer coupling can be realized through the design of the loaded multi-layer FSM structure in the future to widen the working window.

## Figures and Tables

**Figure 1 micromachines-15-00948-f001:**
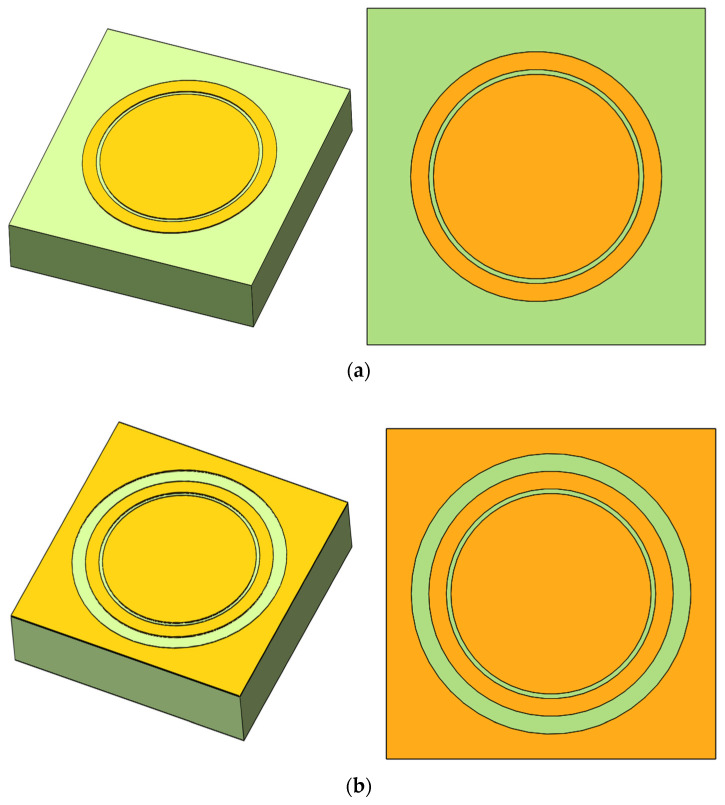
(**a**) Two-dimensional bandstop metal patch. (**b**) Two-dimensional bandpass metal patch.

**Figure 2 micromachines-15-00948-f002:**
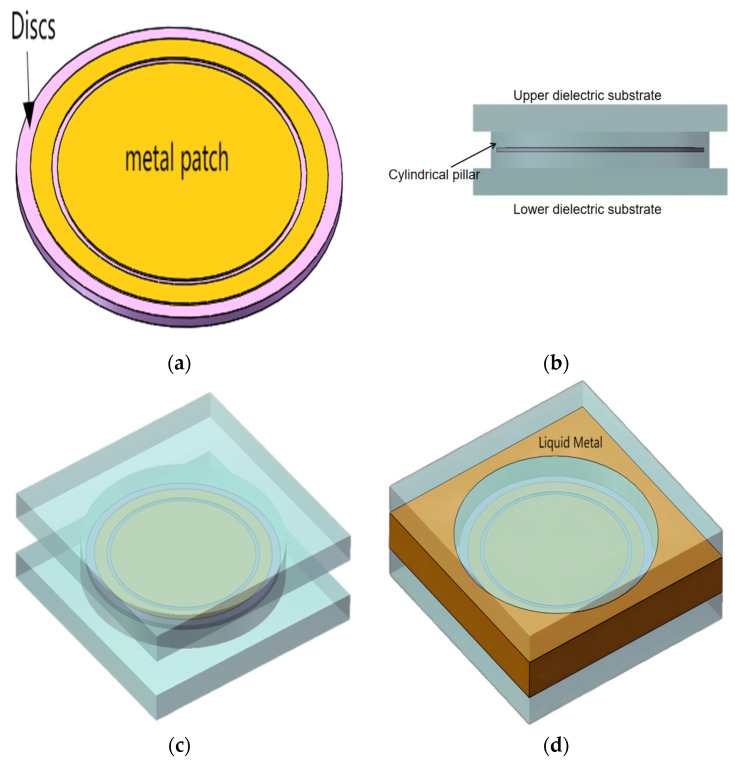
Three-dimensional FSM element structure, (**a**) metallic rings and circular patches are loaded on discs by means of PCBs, (**b**) encapsulation of the structure of (**a**) in the center of a PMMA pillar, (**c**) Case I element structure without liquid metal flow into the cavity structure, and (**d**) Case II element structure with liquid metal flow into the cavity structure.

**Figure 3 micromachines-15-00948-f003:**
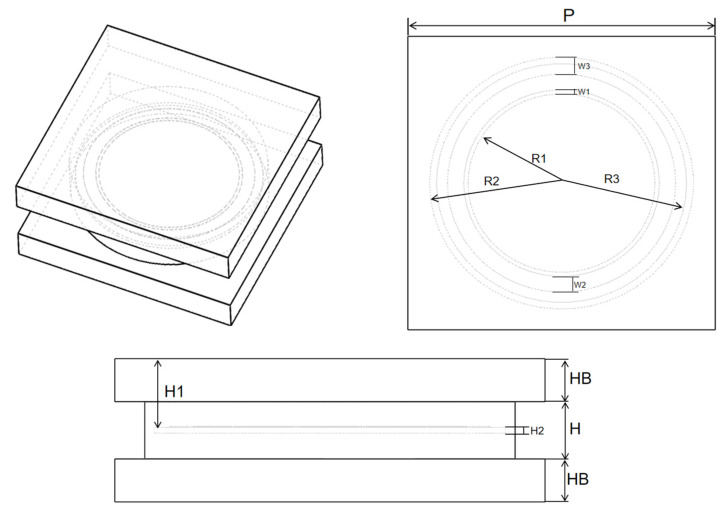
Bandpass/bandstop-switchable FSM element structural parameters.

**Figure 4 micromachines-15-00948-f004:**
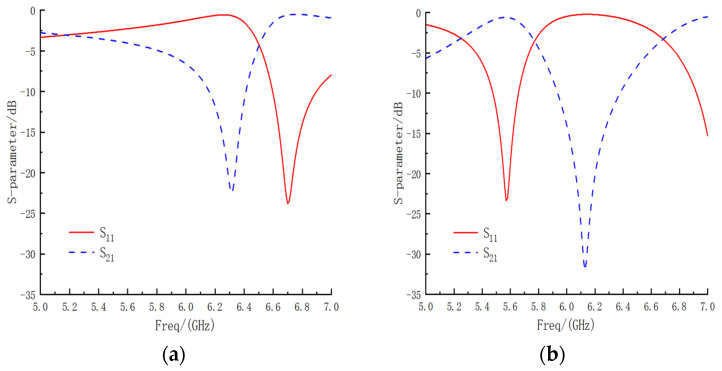
(**a**) S-parameter curve of bandstop (Case I). (**b**) S-parameter curve of bandpass (Case II).

**Figure 5 micromachines-15-00948-f005:**
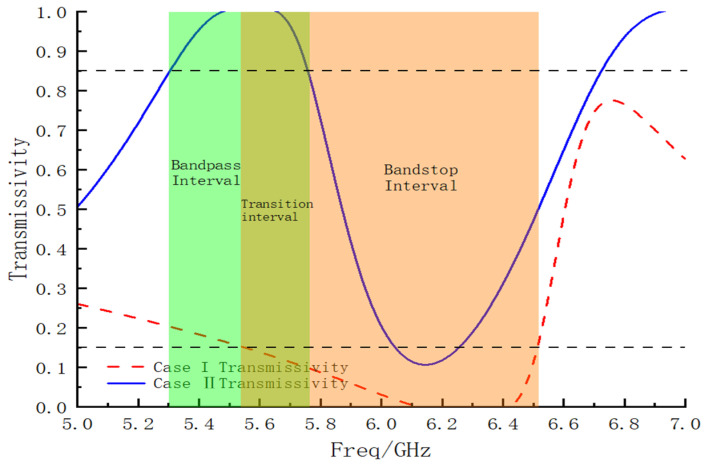
Bandpass/bandstop interval.

**Figure 6 micromachines-15-00948-f006:**
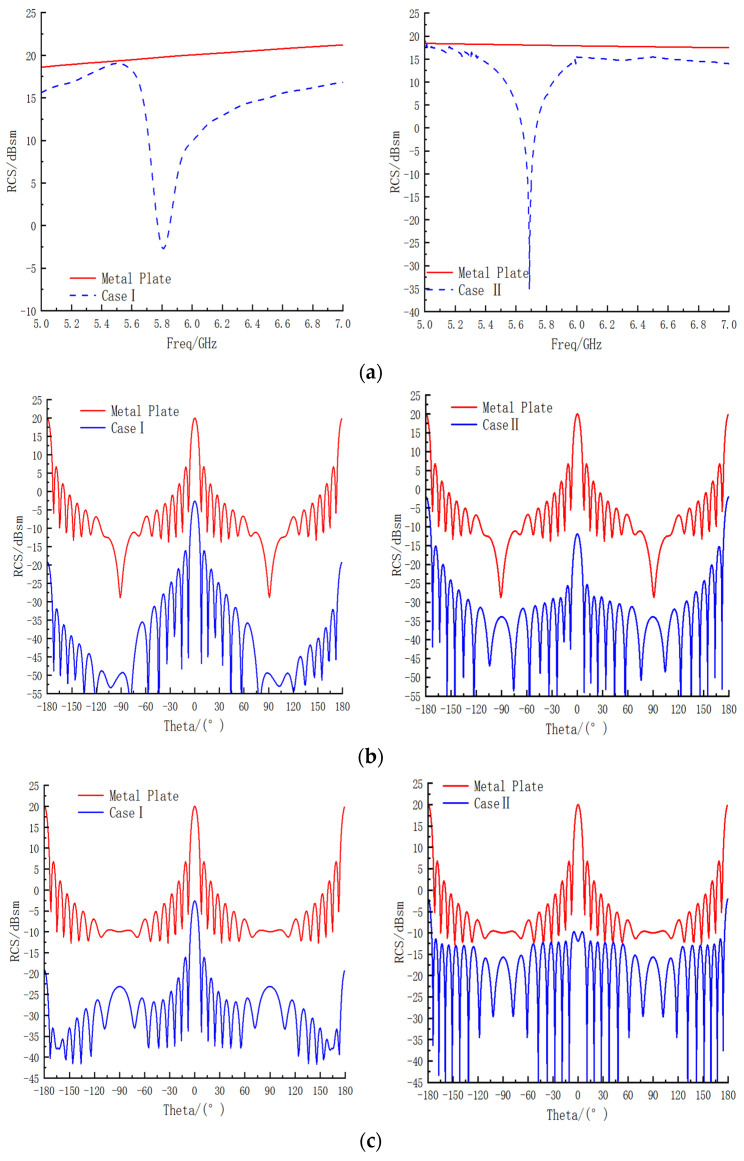
(**a**) RCS frequency curves for Cases I and II. (**b**) RCS with angle curves for TE polarization in Cases I and II. (**c**) RCS with angle curves for TM polarization in Cases I and II.

**Figure 7 micromachines-15-00948-f007:**
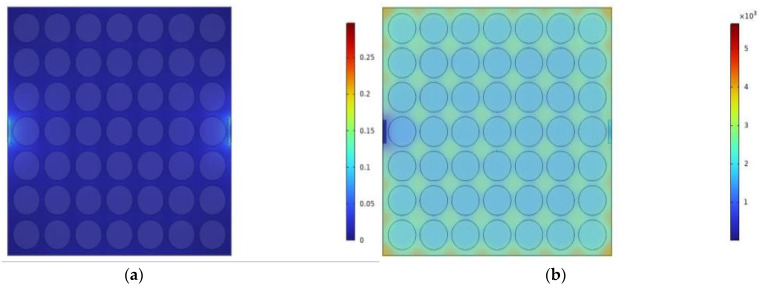
FSM structure flow rate and temperature cloud. (**a**) Air flow chart. (**b**) Air temperature chart. (**c**) LM temperature chart at 0.1 m/s. (**d**) LM temperature chart at 0.2 m/s. (**e**) LM temperature chart at 0.3 m/s. (**f**) LM flow chart at 0.1 m/s. (**g**) LM flow chart at 0.2 m/s. (**h**) LM flow chart at 0.3 m/s.

**Figure 8 micromachines-15-00948-f008:**
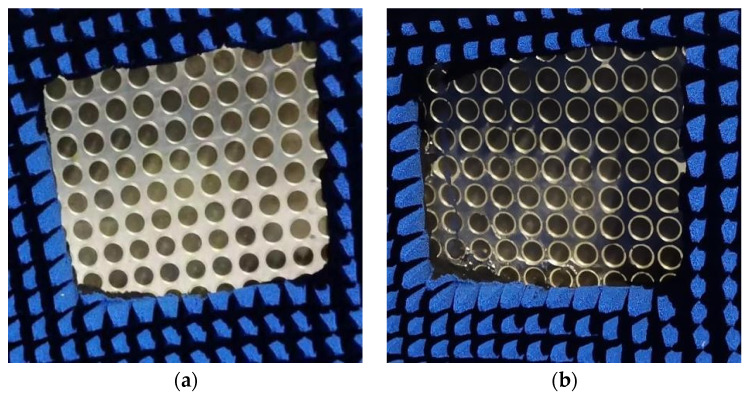
(**a**) LM-withdrawn FSM structure. (**b**) LM-filled FSM structure.

**Figure 9 micromachines-15-00948-f009:**
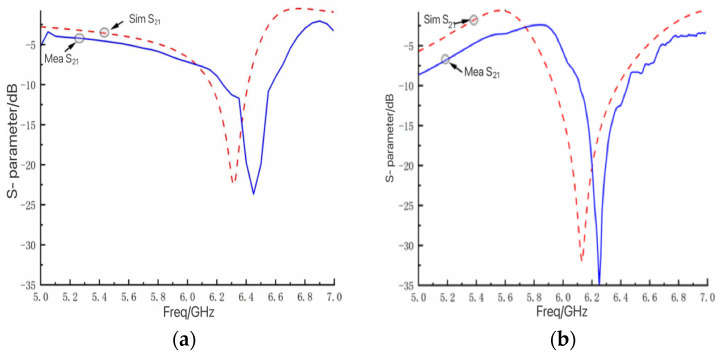
Simulated and measured transmission characteristic curves of the FSM structure. (**a**) Case I and (**b**) Case II.

**Figure 10 micromachines-15-00948-f010:**
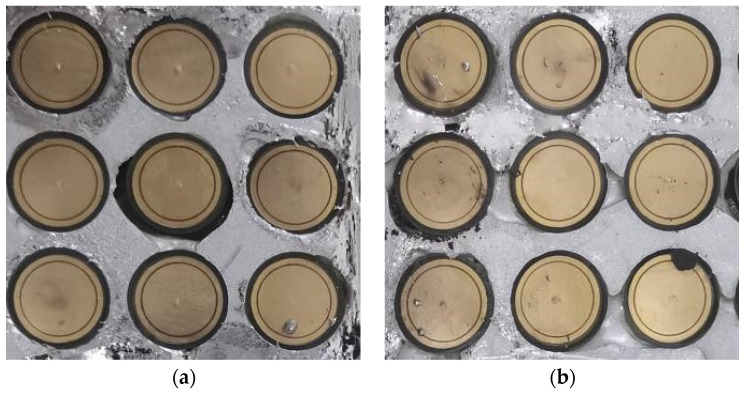
(**a**,**b**) LM-filled FSM structural defect diagram.

**Figure 11 micromachines-15-00948-f011:**
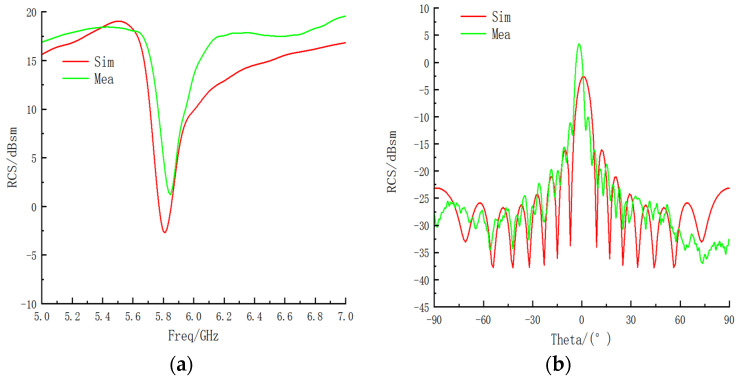
(**a**,**b**) RCS simulation and experimental comparison of the FSM.

**Table 1 micromachines-15-00948-t001:** Related material parameters.

Material	Permittivity	Loss Tangent
Rogers RT/Duroid 5880	2.2	0.0009
PMMA	2.7	0.0078

**Table 2 micromachines-15-00948-t002:** FSM element structural parameters.

Structural Parameters	P	R1	R2	R3	W1	W2	W3	H	H1	H2	HB
Value/mm	15	4.56	6.44	6.1	0.21	0.8	0.8	2	2.5	0.2	1.5

**Table 3 micromachines-15-00948-t003:** RCS-related data sheet of the FSM structure.

		Case I	Case II
Frequency of Extreme Point (GHz)		5.80	5.67
RCS Minimum (dBsm)		−2.571	−17.25
Equal Frequency Metal Plate RCS Value (dBsm)		19.774	19.544
Forward RCS Value (dBsm)	Phi = 0°	−2.798	−12.08
	Phi = 90°	−2.571	−12.38
Lateral RCS Value (dBsm)	Phi = 0°	−49.45	−34.05
	Phi = 90°	−23.120	−33.56

**Table 4 micromachines-15-00948-t004:** Material parameters of thermal analysis simulation.

Material	Densities kg/m^3^	Specific Heat of Mass J/(kg·K)	Heat Conductivity W/(m·K)	Stickiness Pa·s
EgaIn	6363	366	16.5	2.4 × 10^−3^
Air	1.293	1005	0.0261	1.79 × 10^−5^

**Table 5 micromachines-15-00948-t005:** Average and maximum surface temperatures of the FSM structure at different fluid flow rates.

	Air 0.1 m/s	Liquid Metal 0.1 m/s	Liquid Metal 0.2 m/s	Liquid Metal 0.3 m/s
Average temperature/°C	1570.8	91.961	55.625	43.423
Highest temperature/°C	3538.5	211.42	123.28	91.271

**Table 6 micromachines-15-00948-t006:** Comparison of the proposed FSM with existing studies on reconfigurable FSMs.

Ref.	Object of Study	Reconfiguration Category	Control Method	RCS Reduction Performance	Thermal Performance	Frequency/GHz
[26]	Antenna	Polarization	EGaIn	Not mentioned	Not mentioned	2.4
[21]	FSM	Frequency	EGaIn	Not mentioned	1000 °C down to 62 °C	5–13
[23]	FSM	Frequency	Hg	Not mentioned	Not mentioned	4.08–16.96
[24]	FSM	Frequency	Water	Not mentioned	Room temperature (25 °C)	5.2–7.0
[25]	FSM	Frequency	High-voltage current	Not mentioned	Not mentioned	8–12
[27]	FSM	Bandpass/Bandstop	Conduction/disconnection of transmission lines	Not mentioned	Not mentioned	4–10.5
[22]	FSM	Allpass to Bandpass/Bandstop	EGaIn	Not mentioned	Not mentioned	1.36–2.63
This work	FSM	Bandpass/Bandstop	EGaIn	RCS reduction performance was good	1000 °C down to 91.961 °C	5.53–5.76

## Data Availability

The original contributions presented in the study are included in the article, further inquiries can be directed to the corresponding author.

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
