# Peer review of "A High-Temperature-Resistant Stealth Bandpass/Bandstop-Switchable Frequency Selective Metasurface"

_micromachines, 2024, doi:10.3390/mi15080948_

Round 1

Reviewer 1 Report

Comments and Suggestions for Authors

This manuscript presents an innovative design for bandpass/bandstop switchable frequency-selective metasurface, which makes the FSM equipped with high-temperature resistance and low RCS characteristics, and is expected to be applied in high-speed aircraft.

Although the contribution of this work is obvious, and the topic of interest to the Micromachines  readers. But there are still some flaws need to be revised, and make the presentation of this manuscript more clear. They are reported hereafter as they appear in a chronological order within the manuscript:

1. Please clarify the bandpass and bandstop intervals, as well as the switching interval in the abstract.

2. In Sect.IV, consider discussing the significance and potential impact of your findings and pointing out the limitations of your research. Also, provide some specific suggestions or outlooks so that readers can better understand your research results.

3. The author needs to mention some of the latest work on metasurfaces.

4. Please keep the ordinates in Fig.4 (a) (b) and Fig.9 (a) (b) consistent.

Author Response

Comments 1: Please clarify the bandpass and bandstop intervals, as well as the switching interval in the abstract.

Response 1: Thank you for pointing this out. I agree with this comment. Therefore, I have clarify the bandpass and bandstop intervals, as well as the switching interval in the abstract. You can find this change in page 1, line 14-17.

Comments 2: In Sect.IV, consider discussing the significance and potential impact of your findings and pointing out the limitations of your research. Also, provide some specific suggestions or outlooks so that readers can better understand your research results.

Response 2: Agree. I have, accordingly, revised the Sect.IV to emphasize this point you suggested. This change can be found in the line 497-509. 

Comments 3:The author needs to mention some of the latest work on metasurfaces.

Response 3: Thank you for your suggestion.I agree.Accordingly, I have revised the references  to mention some of the latest work on metasurfaces. This change can be found in the page 15. 

Comments 4: Please keep the ordinates in Fig.4 (a) (b) and Fig.9 (a) (b) consistent

Response 4: Thank you for pointing this out. I agree with this comment. I have revised the ordinates in Fig.4(a) (b).

Reviewer 2 Report

Comments and Suggestions for Authors

The manuscript presents a design of a switchable frequency-selective metasurface (FSM) intended for high-temperature applications. While the concept of utilizing liquid metal (LM) for reconfigurability and heat dissipation is technically sound and the reconfigurability between bandpass and bandstop states using LM is interesting, the manuscript does not demonstrate sufficient novelty or significant advancement in the field of frequency-selective surfaces (FSS) and metasurfaces [18, 23]. The authors should highlight new insights or significant improvements over existing designs, such as demonstrating if the insertion loss (S21) and radar cross-section (RCS) reductions are superior to existing technologies. Detailed comments:

(1) Figures 1-3 seem duplicated and not simple and clear enough to understand the structures.

(2) It is strange that the colormaps in Figure 7 (c), (e), and (g) (or Figure 7 (d), (f), and (h)) are the same, and only the values in the color bar differ. 

(3) In the caption of Figure 7, (g) should be LM flow chart and (h) should be LM temperature chart. The authors get the two wrong. 

(4) Please ensure consistent use of terminology: "frequency-selective surface (FSS)" or "frequency-selective metasurface (FSM)".

Comments on the Quality of English Language

(5)  Please proofread thoroughly to correct grammatical errors and improve the overall readability of the manuscript. For example, in the caption of Figure 11, "RCS simulation experimental comparison of the FSM structure" should be "RCS simulation and experimental comparison of the FSM structure".

Author Response

Comments 1: Figures1-3 seem duplicated and not simple and clear enough to understand the structures.

Response 1: Agree. I have revised the Fig.1-3 to description the structures.

Comments 2: It is strange that the colormaps in Figure 7 (c), (e), and (g) (or Figure 7 (d), (f), and (h)) are the same, and only the values in the color bar differ.

Response 2: Thank you for pointing this out.In the previous manuscript, Fig.7(c),(e),(g) did not use a uniform color bar. I have made Fig. 7 use a uniform color bar. As can be seen from these figures, the temperature trends are consistent for all three flow rates. This revision can be found in the Fig.7 

Comments 3: In the caption of Figure 7, (g) should be LM flow chart and (h) should be LM temperature chart. The authors get the two wrong.

Response 3: Thank you for pointing this out.I have revised the caption of Fig.7.

Comments 4: Please ensure consistent use of terminology: "frequency-selective surface (FSS)" or "frequency-selective metasurface (FSM)".

Response 4: Agree. I have ensured that terminology is used consistently.These changes can be found in page 1,2,6,13,14.

Comments 5: Please proofread thoroughly to correct grammatical errors and improve the overall readability of the manuscript. For example, in the caption of Figure 11, "RCS simulation experimental comparison of the FSM structure" should be "RCS simulation and experimental comparison of the FSM.

Response 5: Thank you for pointing this out.I have change it in the line 457.

Reviewer 3 Report

Comments and Suggestions for Authors

In the paper 'A High-Temperature-Resistant Stealth Band-pass/Band-stop Switchable Frequency Selective Metasurface' the authors present their recent results about FSM design. The paper is generally well organized but there are few points that need to be better explained before publication. My major concerns are the following:

1. The final designed structure is not clear at all. In particular Fig.2(d) is not clear. Where are the metallic area? Everything is blue and is not clear what the figure is showing.

2. Why in Fig. 3(b) there are no orange areas? I think the authors should give a clear 3D picture of the final structure indicating all the materials and dimensions. At this stage, I cannot understand the geometry of the final structure.

3. Also, a table of the refractive index for all the material is necessary.

4. I believe that Case I and Case II deserve a picturial/schematic figure.

5. In the text, the authors should explain how they compute the RCS. How it is defined?

6. At line 345 a point is missing before Furthermore.

Author Response

Comments 1: The final designed structure is not clear at all. In particular Fig.2(d) is not clear. Where are the metallic area? Everything is blue and is not clear what the figure is showing.

Response 1: Thank you for pointing this out. I have revised the Fig.2 (d). This change can be found in the Fig.2 and I hope you can understand the final structure.

Comments 2:Why in Fig. 3(b) there are no orange areas? I think the authors should give a clear 3D picture of the final structure indicating all the materials and dimensions. At this stage, I cannot understand the geometry of the final structure.

Response 2: Thank you for your suggestion. I agree with this comment.In Fig.3, I have given a clear 3D picture indicating all the materials and dimensions.This revision can be found in Fig.3.

Comments 3: Also, a table of the refractive index for all the material is necessary.

Response 3: Thank you for pointing this out. I agree with this comment. I've added a table of the refractive index for all the material.You can find this change in line 150.

Comments 4: I believe that Case I and Case II deserve a picturial/schematic figure.

Response 4: Agree. I added the schematic in Fig.2(c)(d).These changes can be found in Fig.2.

Comments 5:In the text, the authors should explain how they compute the RCS. How it is defined?

Response 5: Thank you for your suggestion.I agree with this comment.I've added the RCS definition in line 264-266 and we use CST simulation software to compute the RCS.

Comments 6: At line 345 a point is missing before Furthermore.

Response 6:Thank you for pointing this out. I agree with this comment. I have added a point before Furthermore.This change can be found in line 338.

Round 2

Reviewer 2 Report

Comments and Suggestions for Authors

The authors have resolved my concerns, and I agree with the publication. 

Reviewer 3 Report

Comments and Suggestions for Authors

The authors reply to all my comments